# One-shot-but-not-degraded Federated Learning

## ABSTRACT

Transforming the multi-round vanilla Federated Learning (FL) into one-shot FL (OFL) significantly reduces the communication burden and makes a big leap toward practical deployment. However, we note that existing OFL methods all build on model lossy reconstruction (i.e., aggregating while partially discarding local knowledge in clients' models), which attains one-shot at the cost of degraded inference performance. By identifying the root cause of stressing too much on finding a one-fit-all model, this work proposes a novel one-shot FL framework by embodying each local model as an independent expert and leveraging a Mixture-of-Experts network to maintain all local knowledge intact. A dedicated self-supervised training process is designed to tune the network, where the sample generation is guided by approximating underlying distributions of local data and making distinct predictions among experts. Notably, the framework also fuels FL with flexible, data-free aggregation and heterogeneity tolerance. Experiments on 4 datasets show that the proposed framework maintains the one-shot efficiency, facilitates superior performance compared with 8 OFL baselines (+5.54% on CIFAR-10), and even attains over ×4 performance gain compared with 3 multi-round FL methods, while only requiring less than 85% trainable parameters.

## CCS CONCEPTS

• **Computing methodologies** → **Distributed artificial intelligence**; **Cooperation and coordination**; • **Security and privacy**;

## KEYWORDS

One-shot Federated Learning, Mixture of Experts, Data-free, Heterogeneous Systems

## 1 INTRODUCTION

As a popular machine learning paradigm featured with privacy protection, Federated Learning (FL) enables multiple distributed clients to fuse local knowledge collaboratively without disclosing their raw data [27, 29, 46, 51]. Basically, FL lets clients independently train local models, collects the local knowledge for global aggregation, and distributes the aggregated model for iterative local training and aggregation. However, such multi-round client-server interaction would incur a heavy communication burden (e.g., more than 250 GB for simple VGG19 model [38]) and coordination costs (e.g., client selection for capability alignment [1]), criticized for being prohibitive for real-world implementation [5, 7, 47].

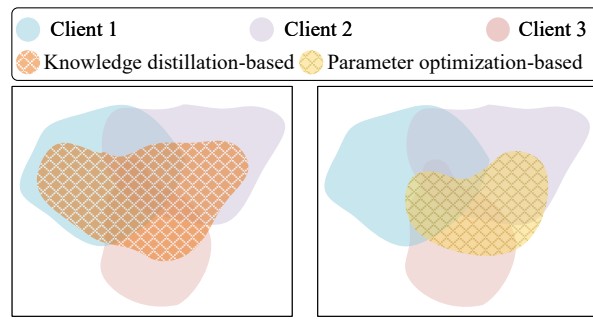

**Figure 1: Visualization of the correct-prediction domains of clients' local models and models learned with existing OFL methods. Large sub-areas of the learned knowledge of local models are discarded with knowledge distillation and parameter optimization.**

As a remedy, one-shot FL (OFL) has emerged recently by reducing the interaction of vanilla FL to just *one communication round* [12]. In fact, OFL is believed to be well-suited for the popular model market scenarios [47], where users are willing to sell local-trained models but are reluctant to join in the redundant training process. Technically, existing designs of OFL, now in its infant age, could be categorized into knowledge distillation methods (KD) [16, 22, 40, 47, 49] and parameter optimization methods [8, 35], both works in a way of aggregating all local models into one global model. Wherein, KD methods transfer knowledge from local models to a global model with assumed-to-be-available auxiliary data, while the latter one focuses on designing dedicated averaging [29, 45], clustering [8], and solving Pareto optimum [35] for one-shot.

Unfortunately, the one-shot gain of existing OFL methods comes at the cost of degraded learning performance. The above proposals all try to learn a one-fit-all global model given local models, which is essentially a model **reconstruction** process, accompanied by knowledge loss. The visualization in Figure 1 shows the differences in correct-prediction domains (i.e., samples that are correctly predicted) between local models and existing OFL models. Notably, both knowledge distillation and parameter optimization methods reconstruct the knowledge space in the union of already-learned local knowledge, while large sub-areas (i.e., those not covered after OFL) are discarded during aggregation. As a result, samples falling in the discarded sub-areas would experience wrong inference with existing OFL models.

The FL paradigm would be more fantastic if the one-shot is attained without performance degradation. For this purpose, this work presents the novel design Intact One-shot Federated Learning (IntactOFL), with the key idea of synergizing the knowledge of local models rather than reconstructing it. By regarding each local model as an expert, IntactOFL employs the Mixture of Experts (MoE) [18] architecture to retain all the local knowledge,

where a gating network is planted for flexible expert/local model assignment during inference (i.e., having which local models to make prediction on a specific sample), thus relieving the one model-fit-all pitfall. With the expert model frozen, the gating network, as the core in our one-shot MoE network, is trained in a self-supervised way based on auxiliary sample generation. Specifically, a generator is designed to construct samples that, on one hand, mimic the local data with the pseudo label approximating the prediction of local experts, and on the other hand, are hard to predict by the MoE network and thus fuel the optimization of it.

We elaborate on and highlight three advantages of IntactOFL: First, in contrast to the previous OFL's reconstruction-based design, IntactOFL preserves all local models' training efforts and utilizes a lightweight gating network to essentially construct a dedicated model for each sample (i.e., **flexible aggregation**). Second, it requires no additional data or pre-trained models (i.e., **data-free**), which well-suits the privacy-sensitive scenarios of FL, especially compared to existing KD methods. Third, by focusing on the fusion of knowledge rather than the aggregation of models, IntactOFL could by nature handle heterogeneous clients with non-iid data distribution and heterogeneous model architectures (i.e., **heterogeneity tolerance**).

Overall, our main contributions can be summarized as follows:

- We first identify the knowledge loss pitfall of existing OFL methods, incurred by the reconstruction of the global model from local models and appeal to synergizing knowledge instead. We believe this could spur rethinking the default way of performing aggregation in FL.
- We invent a novel one-shot learning framework, which attains one-shot with NO performance degradation. Its technical novelty includes adapting the MoE architecture to integrate all local models with no knowledge loss and the design of self-supervised network training based on a distribution-aware and informativeness-sensitive sample generator.
- The proposed IntactOFL is featured with no requirements for additional auxiliary data, pre-trained models, data distribution, and specific model architectures.
- Extensive experiments have evaluated the effectiveness, scalability, and efficiency of IntactOFL, which consistently outperforms 8 OFL baselines on 4 datasets with various heterogeneity settings.

## 2 RELATED WORK

### 2.1 One-shot Federated Learning

One-shot FL is a variant of the FL that requires only one communication round between the server and clients. In OFL, the clients upload well-trained local models to the server, and the server aggregates these uploaded models to obtain a new global model [12]. Existing methods in OFL can be categorized into knowledge distillation-based and parameter optimization-based methods.

*2.1.1 Knowledge Distillation-based.* These methods are proposed to use knowledge distillation (KD) to transfer the massive local knowledge into one global model. They use the local models as teachers, and use the auxiliary public dataset or model to train a student model. [12] firstly proposed to use the KD and use the ensemble prediction of local models as the teacher's output. FedKT [22] designs a two-tier PATE structure [30] relying on public data to improve the ensemble of local models. To alleviate the label skews, FedOV [9] adopts open-set voting in OFL to enhance the generalization ability of the ensemble. However, these methods require the auxiliary public data or pre-trained models should be similar to the original task, which limits the application in real-world scenarios, especially in privacy-sensitive scenarios such as biomedical domains, since there might be no available data or models. To this end, some methods try to utilize additional distilled data or synthetic data instead of public data. [49] proposes transmitting a distilled dataset to the server. DENSE [47] firstly proposes a data-free OFL method through training a data generator to assist the KD process. Co-Boosting [7] designs a mutually enhanced process to synthesize high-quality samples and distillation models. Furthermore, FedCAVE [16] modifies the local learning task into training a conditional variation auto-encoder (CAVE) and uses KD to compress the ensemble into a powerful decoder. The decoder can be used to generate training samples for the global model. And the FedCADO [40] adopts the popular diffusion models to get the synthetic data. However, none of the aforementioned methods can fully use the knowledge of local models, since the KD methods have been criticized as inefficient [48], and existing an intractable performance gap.

*2.1.2 Parameter optimization-based.* These methods aim to search for an optimum across all local models by analyzing the local model parameters. Clearly, the traditional statistic aggregation methods, such as FedAvg, Median [42], and Krum [3] can still apply in OFL but achieve low performance. k-FED [8] runs a variant of Lloyd's method for k-means clustering and obtains an aggregated model through one round iteration of exchanging local cluster means. MA-Echo [35] tries to get the Pareto optimum of the local clients via exploring common harmonized optima. However, directly analyzing the model parameters requires all local models should be homogeneous, whose setting is not practical in real-world heterogeneous scenarios. And the optimum of the parameters can not represent the optimum of the model performance, in most cases, the performance degrades obviously [47]. Besides, none of these methods support the model heterogeneity, *i.e.*, different clients have different model architectures [23]. It is challenging to get the optimum from completely different architecture parameters.

### 2.2 Mixture of Experts

The MoE [6, 18] is an ensemble learning framework that combines multiple expert networks to enhance the overall performance of a model. Each expert in an MoE model is specialized in handling specific types of data or tasks. A gating network, which is a crucial component of MoE, dynamically routes input tokens to the appropriate experts, ensuring that only relevant parts of the model are activated for a given task. Based on this, several improvements have been proposed to reduce the training cost [14, 32], or improve performance on multi-tasks [6, 28].

Recently, MoE has become promising again due to the appearance of Mixtral8×7B, which uses the MoE for model scaling and

**Figure 2: The workflow of IntactOFL. All clients perform local training and upload local models to the server only once. The one-shot MoE network consists of a set of experts and a gating network. The server aggregates these local models into the expert set and trains a gating network in a self-supervised manner. The key steps of each training epoch $t$ are: (1) Given a random noise and pseudo label, the Generator $g$ aims to generate auxiliary data that is similar to the local training data guided by the current MoE network with $\mathcal{L}_{gen}^t$; (2) Using the auxiliary data after augmentation as the input of the MoE network and updating the gating network by $\mathcal{L}_{train}^t$ to form a better MoE network.**

achieves better performance than a larger one. Moreover, GShard [20] uses MoE for scaling up large models and managing models effectively. Switch Transformers [11] introduces a model scaling method by utilizing the MoE to substitute the FNN in Transformer, achieving better training efficiency and model performance. MoE architecture can also be applied to solve the multi-task or multi-modal problem. LLaMA-MoE [36] build a series of MoE models based on LLaMA [37] with continual pre-training. MoE-LLaVA [25] adopt this architecture to achieve high performance and fewer parameters in Large Vision-Language Models (LVLM).

Existing methods also introduce the MoE into FL, and all these methods focus on solving the heterogeneity problem in personalized FL [39]. PFL-MoE [13] views the global model and personalized local model as two experts and utilizes the MoE to achieve better personalized performance and generalization. FedMoE [41] tries to solve the model heterogeneity problem and uses the MoE to weigh the representations of the global homogeneous model and local personalized heterogeneous models. However, these methods still adopt multi-round averaging, which suffers from performance degradation as the parameter optimization-based OFL methods. Contrarily, we adopt the MoE in OFL aggregation, focusing on better leveraging the knowledge of local models.

## 3 FORMULATION AND DESIGN OF INTACTOFL

### 3.1 Learning Problem and Goal

Suppose that we have a set of clients $\mathbb{C}$, with totally $m = |\mathbb{C}|$ clients. Each client $c_k \in \mathbb{C}$ possesses a local private data $\mathbb{D}_k = \{(x_i, y_i)\}_i^{m_k}$, where $m_k = |D_k|$ represents the local data quantity, $x_i$ is the $i^{th}$ sample with the corresponding label $y_i$. The original goal of OFL is to train a single global model $w_g$ over $\mathbb{D} = \cup_{k=1}^m \mathbb{D}_k$ in only one

communication round, which can be described as follows,

$$\min_{w_g} \mathcal{L}(w_g) = \frac{1}{|\mathbb{D}|} \sum_{(x_i, y_i) \in \mathbb{D}} \ell(f(x_i; w_g), y_i), \quad (1)$$

where $\ell$ represents the loss function corresponding to the OFL task, for example, $\ell$ can be the cross-entropy function in the classification task. $f(x_i; w_g)$ is the prediction function that output the prediction of $x_i$ when given the model parameter $w_g$. In OFL, the server can not directly access the data $\mathbb{D} = \cup_{k=1}^m \mathbb{D}_k$, and only well-trained local models $w_k$ are accessed. To facilitate this, existing methods mostly can be categorized into knowledge distillation-based and parameter optimization-based methods.

For knowledge distillation-based methods, the well-trained local models are integrated into an ensemble, and the ensemble model acts as **teacher** to guide the global model (**student**). The ensemble model can be defined as follows,

$$\mathcal{E}(x; \{w_k\}_{k=1}^m) = \sum_{k=1}^m \beta_k f(x; w_k), \quad (2)$$

where $f(x; w_k)$ is the output of the input sample $x$ given the local model $w_k$, while $\beta_k$ is the weight of client $c_k$. In most settings, the $\beta_k$ is assigned to $\beta_k = \frac{1}{m}$ or $\beta_k = m_k / \sum_{j=1}^m m_j$. Relying on the public auxiliary data or synthetic data, i.e., $\mathbb{D}_A$, the distillation process can be defined as follows:

$$\min_{w_g} \mathcal{L}(w_g) = \frac{1}{|\mathbb{D}_A|} \sum_{(x_i, y_i) \in \mathbb{D}_A} \ell(f(x_i; w_g), \mathcal{E}(x_i; \{w_k\}_{k=1}^m)), \quad (3)$$

where the $\ell$ can be the Kullback-Leibler (KL) divergence function corresponding to the distillation task. Notably, existing literature has reported that the performance gap of knowledge distillation [39, 50], the key part is the distillation distance between the ensemble and global model.

For parameter optimization-based methods, the global model $w_g$ is the optimum output across all local models with optimization

mechanism $\mathcal{A}$, that is $w_g = \mathcal{A}(\{w_k\}_{k=1}^m)$. The objective of these methods can be written as:

$$\min_{w_g} \mathcal{L}(\{w_k\}_{k=1}^m) = \min_{w_g} \mathcal{A}(\{w_k\}_{k=1}^m, w_g), \quad (4)$$

where $\mathcal{A}(\{w_k\}_{k=1}^m, w_g)$ can be any distance functions or similarity functions. When $\mathcal{A}(\{w_k\}_{k=1}^m, w_g) = \|\sum_{k=1}^m \rho_k w_k - w_g\|^2$, the global model is $w_g = \sum_{k=1}^m \rho_k w_k$, which is the same as the vanilla aggregation algorithms FedAvg in a one-shot manner. However, these methods focus on processing the model parameter, which is far away from the original OFL objective.

In IntactOFL, we aim to preserve all local models' knowledge through direct integration with MoE. We define the objective of IntactOFL as:

$$\min_{\mathcal{M}} \mathcal{L}(\{w_k\}_{k=1}^m) = \frac{1}{|\mathbb{D}|} \sum_{(x_i, y_i) \in \mathbb{D}} \ell(f_{\mathcal{M}}(x_i; \{w_k\}_{k=1}^m), y_i), \quad (5)$$

where $f_{\mathcal{M}}(x_i; \{w_k\}_{k=1}^m)$ is the output of the MoE network. The goal is to fully utilize all local models and form an MoE network $\mathcal{M}$ which can achieve high performance on predefined tasks. For the MoE network $\mathcal{M}$, it consists of gating network $\mathcal{G}$ and a set of experts $E = \{w_k\}_{k=1}^m$. The MoE architecture dynamically adapts the weight of each expert $\rho_k$ through the gating network $\mathcal{G}$ and maximizes the influence of all experts. The weight can be described as:

$$\rho(x; \mathcal{G}) = softmax(x \cdot \mathcal{G}). \quad (6)$$

For any input sample, the gating network $\mathcal{G}$ will dynamically adjust the weights of the experts and distribute these samples to specialized experts for high performance. We use the weighted outputs of the experts as the final output of the MoE network. To this end, the $f_{\mathcal{M}}(\cdot)$ can be formulated as:

$$f_{\mathcal{M}}(x; \{w_k\}_{k=1}^m; \mathcal{G}) = \sum_{k=1}^m \rho_k(x; \mathcal{G}) f(x; w_k). \quad (7)$$

Therefore, the objective of IntactOFL can be rewritten as

$$\min_{\mathcal{G}} \mathcal{L}_{\mathcal{M}}(\{w_k\}_{k=1}^m) = \frac{1}{|\mathbb{D}|} \sum_{(x_i, y_i) \in \mathbb{D}} \ell(\sum_{k=1}^m \rho_k(x; \mathcal{G}) f(x; w_k), y_i). \quad (8)$$

## 3.2 Framework Overview

The illustration of IntactOFL is shown in Figure 2. After the clients upload their well-pre-trained models, the server aggregates these local models with an MoE architecture. The one-shot MoE network consists of the experts and a gating network. The server treats all the local models as the experts and trains a gating network in a self-supervised manner. Specifically, each training epoch consists of two steps: (1) the current one-shot MoE network guides the data generation process of the Generator $g$ with $\mathcal{L}_{gen}^t$, given random noise, the $g$ can produce many auxiliary samples with similarity and informativeness. (2) Then we add the noise on these samples for data augmentation. These generated samples are treated as the input of the one-shot MoE network. We adopt the $\mathcal{L}_{train}^t$ to update the gating network for forming a better MoE network. Note that the self-supervised MoE training consists of two major processes, which are self-supervised data generation and one-shot MoE network training. The self-supervised data generation is designed for

**data-free training**, and the latter one is for **flexible aggregation**. We present the details of these two processes in § 3.3 and § 3.4, respectively.

## 3.3 Self-supervised Data Generation

Data generation fuels the self-supervised process with optimization basics. For each specific pseudo label, the generator is expected to render noises as the samples with this label. Initially, the MoE's prediction on a generated sample would be very different from the expected pseudo label, which would lead to the optimization of the generator and, more importantly, the gating network before the MoE, via back-propagation. This self-tuning process proceeds until the generator learns how to generate and the gating network learns how to assign weights.

**Similarity Measure.** Basically, the generated samples are settled to mimic the raw local data by representing a similar distribution. We consider the similarity between the generated auxiliary data and the training data. Note that the similarity only focuses on the similar data distribution for utility, rather than visual reconstruction. Following the existing data-free methods, we adopt the Cross-Entropy loss function [7, 47]. Different from existing methods which adopt statistic ensemble output to guide the data generator training, we dynamically use the updated MoE network. The loss for similarity in training epoch $t$ can be formulated as:

$$\mathcal{L}_{sim}^t(\hat{x}, y, f_{\mathcal{M}}^t) = \ell_{CE}(f_{\mathcal{M}}^t(\hat{x}), y), \quad (9)$$

where $\hat{x}$ is the auxiliary data with a corresponding label $y$.

**Informativeness Measure.** Optimized only with the CE loss, the generated auxiliary data can be easily fitted, while may be less representative and contains less information for fine-tuning the backbone gating network. Inspired by [7, 10], the hard samples can embed more representative information compared with easy-to-fit samples. In our context, the samples resulting in more diverse outputs across different experts are considered harder to fit (i.e., more informative in optimizing the gating network). Hence, we use the variance of the experts' output, i.e., $\sigma(f_{\mathcal{M}}^t(\hat{x}))$, as the informativeness of each sample. With this, we could explicitly increase the preference for hard samples in data generation.

By jointly considering similarity and informativeness, the loss function that guides data generator training is formulated as:

$$\mathcal{L}_{gen}^t(\hat{x}, y, f_{\mathcal{M}}^t) = \sigma(f_{\mathcal{M}}^t(\hat{x})) \ell_{CE}(f_{\mathcal{M}}^t(\hat{x}), y), \quad (10)$$

where $\sigma(\cdot)$ is the variance function.

To further promote sample diversity, we introduce a data augmentation module after data generation. Specifically, we further add noise back to the originally generated sample following [7]. As such, we can get more diverse auxiliary samples $\mathbb{D}_A$ of which similarity and informativeness.

## 3.4 One-shot MoE Network

The goal of training the MoE network is to make the best of the knowledge of local data. We use the generated auxiliary data as the training data. To preserve the knowledge of local models without disruption, we froze all experts, since the experts' updating would cause unavoidable forgetting [2]. Instead, we focus on training a high-performance gating network to make the best use of

---

**Algorithm 1** IntactOFL

---

**Input:** Clients' local models $\{w_k\}_{k=1}^m$, auxiliary dataset $\mathbb{D}_A = \emptyset$, generator $g$, gating network $\mathcal{G}$, learning rate of generator and gating network $\eta_g$ and $\eta_{\mathcal{G}}$, generation iterations $T_g$, MoE training epochs $T$, and batch size $b$.

**Output:** MoE network: gating network $\mathcal{G}$ and experts $\{w_k\}_{k=1}^m$.

1: **for** each epoch $t$ to $T$ **do**
2:    // Generate auxiliary data;
3:    Sample a batch of noises and labels $\{z_i, y_i\}_b^{i=1}$;
4:    **for** each $t_g$ to $T_g$ **do**
5:       Generate samples $\{\hat{x}_i\}_b^{i=1} = g(\{z_i\}_b^{i=1})$;
6:       // Update the generator;
7:       $g \leftarrow g - \eta_g \nabla_g \mathcal{L}_{gen}^t$, according to Eq.(10);
8:    **end for**
9:    $\mathbb{D}_A \leftarrow \mathbb{D}_A \cup Augmentation(\{\hat{x}_i\}_b^{i=1})$;
10:   // Update gating network;
11:   **for** each sampling batch $\{x_i\}_b^{i=1}$ in $\mathbb{D}_A$ **do**
12:      $\mathcal{G} \leftarrow \mathcal{G} - \eta_{\mathcal{G}} \nabla_{\mathcal{G}} \mathcal{L}_{train}(\mathcal{G})$, according to Eq.(13).
13:   **end for**
14: **end for**

---

the knowledge of local models. To this end, we update the gating network according to Eq.(11).

$$\min_{\mathcal{G}} \mathcal{L}_{\mathcal{M}}(\{w_k\}_{k=1}^m) = \frac{1}{|\mathbb{D}_A|} \sum_{(x_i,y_i)\in\mathbb{D}_A} \ell(\sum_{k=1}^m \rho_k(x;\mathcal{G})f(x;w_k), y_i). \tag{11}$$

Meanwhile, the 'over-trust' problem of the gating network has been widely reported [20, 33], that is, the gating network always produces large weights for the same few experts. Following [33], we add a balance loss to mitigate the weight bias of the gating network. Specifically, we compute the importance of each expert in every batch in $t$ iteration, and the balance loss equals the square of the coefficient of variation ($C\mathcal{V}$) of the set of importance (see Eq.(12)). $\lambda$ is the scaling factor for the balance loss, which is set as 1 in the implementations.

$$\mathcal{V}^t(X) = \sum_{x\in X} \rho^t(x;\mathcal{G}),$$
$$\mathcal{L}_{balance}^t(X) = \lambda \cdot C\mathcal{V}(\mathcal{V}^t(X))^2. \tag{12}$$

In summary, the MoE training loss function consists of two parts, the first part is for high performance which minimizes the difference between model outputs and labels; the second part is for preventing the "over-trust" phenomenon and achieving a balanced gating network. We conclude the loss function as:

$$\min_{\mathcal{G}} \mathcal{L}_{train}^t(\{w_k\}_{k=1}^m) = \mathcal{L}_{\mathcal{M}}^t(\{w_k\}_{k=1}^m; \mathbb{D}_A) + \mathcal{L}_{balance}^t(\mathbb{D}_A). \tag{13}$$

The details of the training process are presented in Algorithm 1.

## 4 EVALUATION

### 4.1 Experiment Setup

We provide the important details of the datasets, data partition, and baselines here, and present the rest details in the Appendix.

*4.1.1 Datasets.* We evaluate the proposed IntactOFL and baselines on four widely used classification benchmarks: CIFAR-10, CIFAR-100, SVHN, and Tiny-ImageNet. CIFAR-10 consists of 60,000 32 $\times$ 32 RGB images in 10 categories. It has 50,000 training samples and 10,000 test samples. CIFAR-100 has the same format as CIFAR-10, except it has 100 categories. SVHN is a real-world dataset comprising over 600,000 digit images extracted from Google Street View images. Tiny-ImageNet is the subset of the ImageNet dataset, containing 200 categories and 100,000 images (500 images for each category), each image is downsized to 64 $\times$ 64.

*4.1.2 Data Partition.* To simulate the real-world heterogeneous environment, we adopt the Dirichlet distribution ($\alpha$) to control the proportions of each category across clients [21, 44]. We use the $\alpha$ to control the degree of heterogeneity. A small $\alpha$ represents a biased data distribution. Following the settings in [47], we set $\alpha = \{0.05, 0.1, 0.3, 0.5\}$.

*4.1.3 Baselines.* We compare the performance of the proposed IntactOFL against the existing two categories of methods. One is the knowledge distillation-based method, another is the parameter optimization-based method. We compare with knowledge distillation-based SoTA which are DENSE [47] and Co-Boosting [7]. Meanwhile, following the setting in [7, 47], we also compare with the FedDF [26] which uses the real validation dataset for distillation, and some baselines derived from prevailing data-free knowledge distillation methods which are F-ADI [43] and F-DAFL [4]. For parameter optimization-based methods, we choose high-performance MA-Echo [35] and vanilla FedAvg in a one-shot manner [12, 29]. To ensure fair comparisons, we omit the comparison with methods that require multi-round interactions, such as FedProx [23], SCAFFOLD [19], and FedCav [45]. We also neglect some methods that require additional public data or models, such as FedKT [22], FedOV [9] and FedGen [50].

## 4.2 Main Results

*4.2.1 Effectiveness.* Table 1 shows the effectiveness of the proposed IntactOFL, we conduct experiments under various data heterogeneity settings across different datasets and methods by vary $\alpha = \{0.05, 0.1, 0.3, 0.5\}$. From the table, we conclude that the proposed IntactOFL is effective in data heterogeneity settings. Specifically, **(a)** In all data heterogeneity settings, the IntactOFL outperforms than baselines. The IntactOFL surpasses the best baseline by substantial margins with 5.54%, 4.26%, 3.50%, and 2.82% on CIFAR-10, CIFAR-100, SVHN, and Tiny-ImageNet, respectively. Even in a more heterogeneous environment ($\alpha = 0.05$), IntactOFL achieves over a 5% accuracy improvement compared to the best baseline. **(b)** Notably, the knowledge distillation-based baselines achieve better performance than the parameter optimization-based methods. The DENSE and Co-Boosting both achieve $\sim$3% test accuracy higher than MA-Echo and FedAvg. The reason is that the parameter reconstruction process of optimization-based methods does not preserve the local models' knowledge, and the knowledge distillation-based methods can transfer this knowledge to a new model. Owing to the MoE architecture which preserves the local models, the proposed IntactOFL can achieve better performance among these baselines. **(c)** Besides, the Ensemble which equally averages all local models'

**Table 1: Performance on four data heterogeneity (varying $\alpha = \{0.05, 0.1, 0.3, 0.5\}$, lower $\alpha$ represents more heterogeneous) on four benchmarks (CIFAR-10, CIFAR-100, SVHN, and Tiny-ImageNet). Underline/bold fonts highlight the best baseline/the proposed IntactOFL.**

| Method | $\alpha$ | MA-Echo | FedAvg | FedDF | F-ADI | F-DAFL | Ensemble | DENSE | Co-Boosting | Ours |
|---|---|---|---|---|---|---|---|---|---|---|
| CIFAR-10 | 0.05 | 36.77±0.91 | 12.13±2.11 | 35.53±0.67 | 35.93±1.56 | 38.32±1.40 | 41.36±0.67 | 38.37±1.08 | 39.20±0.81 | **48.22±0.43** |
| | 0.1 | 51.23±0.28 | 17.43±0.51 | 41.58±0.80 | 48.35±1.23 | 46.34±1.12 | 45.43±0.32 | 50.26±0.24 | 58.49±1.24 | **61.13±0.63** |
| | 0.3 | 60.14±0.21 | 28.07±0.89 | 44.78±0.60 | 52.66±1.44 | 54.03±1.71 | 62.18±0.34 | 59.76±0.45 | 67.21±1.76 | **70.21±0.60** |
| | 0.5 | 64.21±0.23 | 35.42±0.67 | 54.58±0.73 | 58.78±1.67 | 59.09±2.23 | 61.61±0.23 | 62.19±0.12 | 70.24±2.34 | **79.93±0.23** |
| CIFAR-100 | 0.05 | 19.54±0.45 | 4.77±0.21 | 15.07±0.74 | 14.65±0.98 | 16.31±0.33 | 20.46±0.62 | 18.37±2.43 | 20.19±1.44 | **27.99±0.67** |
| | 0.1 | 29.11±0.26 | 6.45±0.71 | 27.17±0.55 | 28.13±1.24 | 26.80±1.33 | 26.23±0.55 | 32.03±0.44 | 27.59±1.35 | **39.15±0.46** |
| | 0.3 | 37.77±0.24 | 10.67±0.31 | 31.23±0.79 | 33.18±0.67 | 34.89±1.45 | 38.01±0.67 | 37.33±0.48 | 39.30±1.30 | **41.86±0.60** |
| | 0.5 | 41.94±0.21 | 12.13±0.05 | 35.39±0.47 | 39.44±1.11 | 37.88±1.34 | 41.61±0.77 | 38.84±0.39 | 42.67±1.40 | **46.78±0.78** |
| SVHN | 0.05 | 44.18±0.34 | 19.43±2.44 | 48.35±0.52 | 47.12±1.34 | 48.75±1.88 | 52.34±171 | 47.28±1.11 | 53.45±1.46 | **59.88±0.89** |
| | 0.1 | 56.33±0.25 | 36.77±0.71 | 49.34±0.57 | 52.67±1.22 | 52.46±1.24 | 57.44±0.35 | 55.28±0.56 | 62.36±1.65 | **63.23±0.12** |
| | 0.3 | 79.94±0.41 | 49.25±0.24 | 63.90±0.39 | 66.50±1.67 | 64.32±1.88 | 79.86±0.21 | 79.43±0.58 | 79.99±2.03 | **83.22±0.06** |
| | 0.5 | 80.23±0.24 | 57.61±0.75 | 72.11±0.47 | 77.62±2.01 | 74.55±1.80 | 81.22±0.17 | 80.03±0.24 | 81.34±1.03 | **84.81±0.12** |
| Tiny-ImageNet | 0.05 | 15.46±0.66 | 5.67±0.45 | 11.45±0.40 | 13.92±1.99 | 15.12±1.34 | 13.28±0.67 | 18.77±0.67 | 19.00±1.45 | **20.45±0.34** |
| | 0.1 | 22.23±0.56 | 8.31±0.21 | 16.32±0.33 | 19.00±1.78 | 19.01±1.11 | 15.38±0.23 | 22.25±0.33 | 21.90±1.20 | **28.43±0.17** |
| | 0.3 | 23.46±0.19 | 13.61±0.10 | 17.79±0.57 | 26.01±1.44 | 23.78±1.23 | 17.53±0.31 | 28.14±0.34 | 29.24±1.32 | **30.15±0.12** |
| | 0.5 | 28.21±0.42 | 13.71±0.16 | 27.55±0.66 | 29.98±1.34 | 27.98±1.10 | 28.50±0.46 | 32.34±0.32 | 30.78±2.01 | **35.09±0.14** |

output can achieve the second-best performance in some settings. This is thanks to the benefits of preserving all information. However, the compromised results obtained by equally averaging limits its performance in all settings. The proposed IntactOFL adopts the gating network for effective utilization of local models' knowledge evidently improves the entire performance. In summary, **the IntactOFL is effective in various data heterogeneity scenarios and achieves competitive performance than baselines.**

*4.2.2 Scalability.* We evaluate the scalability of IntactOFL in two aspects: **horizontal scalability**, which evaluates the performance in a larger distributed network with more clients; and **vertical scalability**, which evaluates the performance in scenarios of model heterogeneity among clients, where different clients can possess entirely distinct model architectures.

For horizontal scalability, we evaluate the test accuracy of all methods in diverse networks by varying the number of clients $m = \{5, 10, 25, 50, 100\}$ on CIFAR-10 and SVHN. Table 2 shows the results of different methods across different clients $m$ in CIFAR-10, the results of SVHN are presented in the Appendix. As suggested in [24], the server can become a major bottleneck while the number of clients increases. We also reach a similar conclusion, with the number of clients $m$ increasing, the performance decreases, which is consistent with [7, 24, 47]. Even though the negative impact on the accuracy, our methods still achieve better performance than other baselines, which also verifies that the knowledge-preserved method brings a better aggregated global model. In summary, the IntactOFL is scalable across diverse distributed networks of varying sizes.

For vertical scalability, we note that our proposed methods can support heterogeneous models, which are scalable to different model architectures. We set five different model architectures from simple to complex, that is, MLP, CNN, MobileNetV2 (Mob) [31],

ResNet (Res) [15], and VGG [34]. We evaluate the test accuracy of all methods with the same setting in § 4.2.1 except $\alpha = 0.5$. Besides, we consider a challenging setting in FL, in which both the data and models are heterogeneous across all clients. Different clients can train their own models on their local data with entirely different model architectures. We remark that this practical setting is complicated in existing methods [7, 17, 47]. Thus, we conduct two types of evaluations to verify the ability to support heterogeneous models. Since some baselines do not support the model heterogeneity, we omit these methods and only report the Ensemble, DENSE, and Co-Boosting.

For the first type, we evaluate the performance by varying the model architectures of entire systems. All clients are homogeneous and test on different model architectures. Results are demonstrated on the left panel of Figure 3. We conclude that the model architecture can impact the performance, a model with better representation ability can achieve better performance. The proposed method achieves the best performance across all model architectures.

The second type concerns a scenario where the model architectures among clients are completely heterogeneous. We set three cases of these complete model heterogeneity, see the central panel in Figure 3. For example, the *Case #1* represents that Client 0 uses MLP to train a local model with the test accuracy of 39.92%, after the local training, the server would collect the MLP from Client 0, VGG from Client 1, ResNet from Client 2, CNN from Client 3, and MobileNet from Client 4. We report the test accuracy of the global model, which is aggregated by completely heterogeneous local models. The results (see the right panel in Figure 3) demonstrate that even in complete model heterogeneity settings, our methods can achieve ~10% accuracy improvement than existing baselines. In summary, our proposed method is **heterogeneity-tolerant** and significantly outperforms other baselines.

**Table 2: Test accuracy of the server model on CIFAR-10 across different numbers of clients** $m = \{5, 10, 25, 50, 100\}$**.**

| $m$ | MA-Echo | FedAvg | FedDF | F-ADI | F-DAFL | Ensemble | DENSE | Co-Boosting | Ours |
|---|---|---|---|---|---|---|---|---|---|
| 5 | 64.21 | 35.42 | 54.58 | 59.34 | 58.59 | 61.61 | 62.19 | 55.34 | **79.93** |
| 10 | 52.64 | 32.09 | 48.88 | 46.33 | 45.45 | 60.44 | 54.67 | 51.11 | **69.11** |
| 25 | 48.36 | 28.03 | 35.44 | 31.83 | 32.88 | 58.44 | 49.32 | 49.32 | **64.32** |
| 50 | 45.35 | 28.24 | 29.91 | 27.66 | 29.98 | 52.51 | 48.67 | 44.56 | **59.45** |
| 100 | 38.54 | 27.14 | 25.66 | 24.89 | 28.91 | 45.72 | 43.34 | 42.45 | **53.21** |

In summary, **the proposed IntactOFL is scalable both in large distributed network and model heterogeneity scenarios and outperforms other baselines by a large margin.**

*4.2.3 Efficiency.* Note that the proposed IntactOFL is efficient. We evaluate the efficiency from two aspects: parameter efficiency and computation efficiency. The parameter efficiency means the required parameters should be as small as possible. The computation efficiency means the training iterations should be as few as possible. We adopt the same setting as the § 4.2.1.

For parameter efficiency, we report the # trainable parameters and the test accuracy of DENSE and Co-Boosting across different global model architectures. The trainable parameters of DENSE and Co-Boosting mainly include the data generator and student model for distillation. In IntactOFL, there is no additional global model, the trainable part only consists of the generator and lightweight gating network. As shown in Figure 4, **thanks to the lightweight gating network, the proposed IntactOFL achieves the highest accuracy while requiring the smallest trainable parameters (85% of DENSE(CNN)) on the server**.

Notably, the IntactOFL is also computation-efficient. We consider that the computation of the local training is essential to learn the local knowledge. However, the aggregation process is inefficient, so existing methods introduce additional computation overhead to distill local models or require multi-rounds of local training. To better evaluate the computation overhead, we design a new factor named Performance Gain (PG), which can be formulated as:

$$PG = \frac{P}{C_{server} + C_{clients}}, \tag{14}$$

where $P$ is the performance, and in the classification task, $P$ is the test accuracy; $C_{server}$ is the computation cost on the server, including training the generator, distilling, and so on. $C_{clients}$ is the sum of local training iterations of all clients. Here, We use a coarse-grained estimation of the number of iterations $T$ to approximate the computational cost, which is $C_{server} \approx T_{server}$ and $C_{clients} \approx T_m \times \sum_{k=1}^{m} E_i$, where $T_m$ is the interaction rounds between the server and clients. For OFL, $T_m = 1$, for multi-rounds FL, $T_m > 1$. **We use the PG to evaluate how much the performance gain is brought by each training iteration.**

Furthermore, we introduce some representative multi-rounds federated learning methods for comparison, which are multi-rounds FedAvg (M-FedAvg) [29], SCAFFOLD [19], and FedCav [45]. As reported in Table 3, we achieve the highest *PG* across all baselines, which means that our methods can obtain the highest global model accuracy improvement for each training step. It is worth noting that, even though the multi-rounds FL algorithms can achieve the

**Table 3: The Performance Gain (PG) of all methods on CIFAR-10 with data heterogeneity** $Dirichlet(\alpha = 0.5)$**.** $T_X$ **represents the total iterations on the server corresponding to method** $X$**.**

| Methods | $C_{server} + C_{clients}$ | P | $PG\ (\times 10^{-2})$ |
|---|---|---|---|
| MA-Echo | $T_{MA-Echo} + \sum_{k=1}^{m} E_i$ | 64.21 | 2.7917 |
| FedAvg | $\sum_{k=1}^{m} E_i$ | 35.42 | 1.7710 |
| FedDF | $T_{DF} + \sum_{k=1}^{m} E_i$ | 54.58 | 2.3730 |
| F-ADI | $T_{ADI} + \sum_{k=1}^{m} E_i$ | 59.34 | 2.5800 |
| F-DAFL | $T_{DAFL} + \sum_{k=1}^{m} E_i$ | 58.59 | 2.5474 |
| Ensemble | $\sum_{k=1}^{m} E_i$ | 61.61 | 3.0805 |
| DENSE | $T_{DENSE} + \sum_{k=1}^{m} E_i$ | 62.19 | 2.7039 |
| Co-Boosting | $T_{Co-Boosting} + \sum_{k=1}^{m} E_i$ | 55.34 | 2.4061 |
| M-FedAvg | $T_m \sum_{k=1}^{m} E_i$ | 47.62 | 0.4762 |
| SCAFFOLD | $T_m \sum_{k=1}^{m} E_i$ | **82.59** | 0.8259 |
| FedCav | $T_m \sum_{k=1}^{m} E_i$ | 81.18 | 0.8118 |
| Ours | $T_{Ours} + \sum_{k=1}^{m} E_i$ | 79.93 | **3.4752** |

best performance, its *PG* is significantly less than that of the OFL, which also verifies the inefficiency of the existing multi-rounds aggregation methods [47].

In summary, **the proposed IntactOFL is parameter-efficient and computation-efficient compared with other baselines**, which requires less trainable parameters and fewer computation iterations while achieving higher performance.

## 4.3 Analysis of the IntactOFL

*4.3.1 Impact of MoE Architectures.* In this part, we investigate the impact of different MoE architectures. Since the clients determine the local models' (experts) architecture, we focus on the performance by varying different gating networks. *Softmax* gating, noisy Top-K gating [32], and MMoE [28] are three popular gating networks in MoE. The *Softmax* gating uses the softmax function to normalize the weights across all experts. The noisy Top-K gating adds noise $\mathcal{N} \sim (0, \sigma)$ to the original weights and only selects top k experts. MMoE adopts multiple gating networks for multi-tasks and here we derive this method by averaging the outputs of these $K_g$ gating networks as the final output. Here, we use the averaged memory consumption of one batch of samples to represent the computational cost. We test these methods on the CIFAR-10 with CNN among five clients. We report the performance and memory consumption of different gating networks in Figure 5. **(a)** We notice that the test accuracy and memory consumption act as a trade-off, utilizing more experts can achieve high performance meanwhile

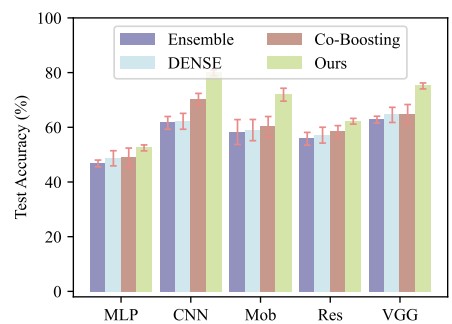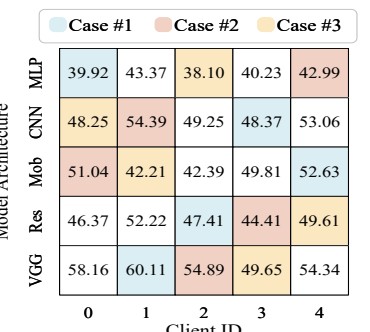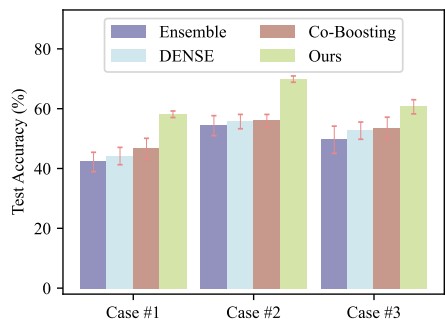

Figure 3: Left panel: Performance on different model architectures. Central panel: Visualization of the local model performance with different architectures. Right panel: Performance on three completely model heterogeneity cases.

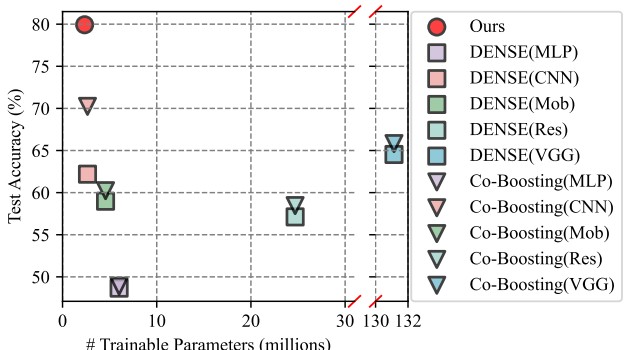

Figure 4: Test accuracy v.s. # trainable parameters on CIFAR-10. We compare the IntactOFL with DENSE and Co-Boosting by varying their global model architectures such as MLP, CNN, MobileNet, ResNet, and VGG.

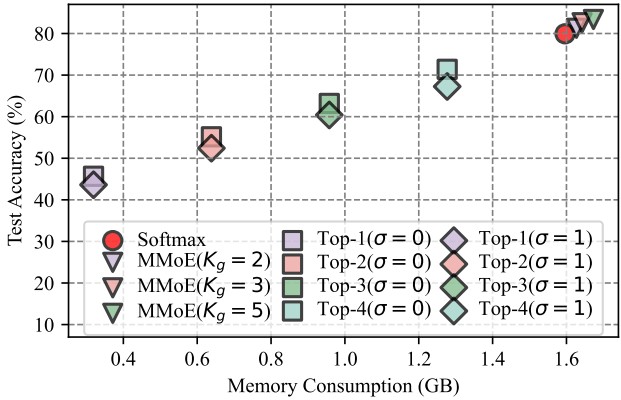

Figure 5: Test accuracy and memory consumption of different gating network architectures on CIFAR-10, such as *Softmax,* noisy Top-K, and MMoE. We vary the number of gating networks ($K_g$) in MMoE and the topk experts ($K = \{1, 2, 3, 4\}$) with the noise level $\sigma = \{0, 1\}$.

**Table 4: Impact of the key components in data generation.**

|  | CIFAR-10 | CIFAR-100 | SVHN | Tiny-ImageNet |
|---|---|---|---|---|
| Ours | **79.93** | **46.78** | **84.81** | **35.09** |
| w/o *Aug* | 76.41 | 44.45 | 83.79 | 34.46 |
| w/o *Inf* | 72.68 | 43.69 | 80.66 | 31.91 |
| w/ *Sim* | 53.12 | 36.47 | 73.11 | 17.98 |

causing larger memory consumption. **(b)** The noise is designed to make MoE training more balanced. However, too much noise can harm the model's performance. **(c)** Using more gating networks for performance increment has shown limited improvement.

*4.3.2 Impact of Generator.* We provide the visualization of the generated data on CIFAR-10 and SVHN in the Appendix. Note that the auxiliary data are generated for utilization rather than visual reconstruction. Besides, we investigate the impact of the key components in data generation. We conduct leave-one-out testing and report the results by removing data augmentation (w/o *Aug*), removing informativeness (w/o *Inf*), and removing both (w/ *Sim*). As shown in Table 4, only considering the similarity leads to poor performance, which is consistent with [7, 47]. The informativeness module is essential for performance improvement compared with the data augmentation. A combination of these components leads to a high performance, which shows that all these components have contributed to performance improvement.

## 5 CONCLUSION

In this paper, we aimed to bridge the knowledge loss of existing one-shot federated learning methods, where the model reconstruction process results in significant performance degradation. We proposed IntactOFL, a novel method that adopts the Mixture of Experts' architecture to preserve all local models' knowledge, achieving high performance through dynamic weighting by a gating network. We designed a self-supervised MoE training framework by iteratively generating samples and updating the gating network. Extensive experiments have verified the effectiveness, scalability, and efficiency of the proposed IntactOFL.

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
