# OpenReview forum: "One-shot-but-not-degraded Federated Learning"
_acmmm.org/ACMMM/2024/Conference — MM2024 Poster_

### Official Review · Reviewer_w5Xt · 2024-05-24

**Rating:** 4
**Confidence:** 3

**Summary:**

This paper introduces a novel one-shot federated learning algorithm, IntactOFL. The algorithm uses a MoE network to maintain the integrity of all local knowledge. By using the gating network of the MoE to integrate local models and alternating the training of an auxiliary data generator and the gating network, it achieves one-shot federated learning through the generation of auxiliary data. Experimental results demonstrate that this federated learning approach achieves one-shot communication without any performance loss and even outperforms traditional federated learning methods.

**Strengths:**

The proposed approach, IntactOFL, is bold and innovative. I like the idea of integrating a synthetic data generator into MoE for training a global model. The paper is well-written and clear in both its problem formulation and algorithm description. Additionally, through experimental study, IntactOFL outperforms other baselines on various datasets, validating the effectiveness of one-shot federated learning in IntactOFL without any performance loss. The visualizations of the auxiliary data points in the supplementary material demonstrate the privacy capabilities of IntactOFL. Furthermore, the approach can be applied to scenarios where the local models across clients differ.

**Limitations:**

1. Figure 1 lacks sufficient insight and could be improved to better illustrate the concept. The statement in Figure 1 should be clarified with additional supporting analysis.

2. I am curious about the optimal size of the generated auxiliary data for achieving good performance. How large should the auxiliary dataset be? The size of the auxiliary dataset will impact communication costs. Although one-shot federated learning only requires one round communication, the single communication load for transmitting the auxiliary dataset is significant. Additionally, the size of the auxiliary dataset could affect the final prediction results. Therefore, a detailed discussion and guidelines on the optimal size of the auxiliary data are necessary.

3. The IntactOFL method proposed in the paper seems to require significant computational resources. The process involves training the auxiliary data generator and the gating network successively in a loop, which may demand extensive computational power. Along with the discussion of this limitation, the paper should provide some potential optimization methods or directions to address this issue. Otherwise, the proposed one-shot federated learning method could be limited in edge scenarios with restricted computational capabilities.

4. Some minor points:

    4a) More categories of one-shot federated learning are missing here, such as Dataset-distillation-based one-shot federated learning.

    4b) The visualization of auxiliary data (now in supplementary material) should be better included in the main paper, as the generated data for the gating network is a crucial part of the proposed framework.

    4c) Although it might be somewhat difficult, providing more theoretical analysis regarding the convergence of the proposed training process would be beneficial.

    4d) It is recommended to provide open-source code for the replication of the experimental results.

    4e) The notations $m$ and $m_k$ are a bit confusing. I suggest using different symbols to distinguish between the number of clients and the number of data points.

**Suitability:**

3

---

### Official Review · Reviewer_NtEr · 2024-05-25

**Rating:** 4
**Confidence:** 4

**Summary:**

Existing OFL approaches all build on model lossy reconstruction, which attains one-shot at the cost of degraded inference performance.
To address this issue, this paper proposes a novel one-shot FL framework by embodying each local models as an independent expert
and leveraging a MoE network to maintain all local knowledge intact. A dedicated self-supervised training process is designed to train the MoE network, where the sample generation is guided by approximating underlying distributions of local data and making distinct predictions among experts. Extensive experiments demonstrate the superiority of the IntactOFL method.

**Strengths:**

1. The motivation for this work is clear.
2. Unlike the common knowledge distillation methods and parameter optimization methods, the authors use the MoE network for OFL.
And treat each client as an expert.
3. The authors utilized auxiliary data generated by the generator to train the MoE network.
4. The authors have given enough consideration to experiments, and the paper has a relatively large number of types of experiments.

**Limitations:**

1. The authors claim that IntactOFL is compared to 8 OFL baselines. But it is always known that vanilla FedAvg is a method that requires
multi-round interactions, not a OFL method.

2. Whether it would be better to train Gating network directly using proxy datasets (some KD methods also utilize proxy datasets),
 I'm not sure if the authors have considered.

> 1.Yi L, Yu H, Ren C, et al. FedMoE: Data-Level Personalization with Mixture of Experts for Model-Heterogeneous Personalized Federated Learning[J]. arXiv preprint arXiv:2402.01350, 2024.

> 2.Parsaeefard S, Etesami S E, Garcia A L. Robust federated learning by mixture of experts[J]. arXiv preprint arXiv:2104.11700, 2021.

> 3.Guo B, Mei Y, Xiao D, et al. PFL-MoE: Personalized federated learning based on mixture of experts[C]//Web and Big Data: 5th International Joint Conference, APWeb-WAIM 2021, Guangzhou, China, August 23–25, 2021, Proceedings, Part I 5. Springer International Publishing, 2021: 480-486.

**Suitability:**

2

---

### Official Review · Reviewer_AWD8 · 2024-05-28

**Rating:** 3
**Confidence:** 3

**Summary:**

This work aims to promote the model's performance under the one-shot FL scenario. To this end, the authors propose a novel strategy to leverage a MioE network to maintain knowledge from local clients. Some experiments are conducted to verify the effectiveness of the proposed method.

**Strengths:**

The proposed method is technically sound. Moreover, leveraging self-supervised data generation is novel in one-shot FL scenarios.

The performance gain is decent. The ablation studies conducted provide a clear picture of the proposed method's effectiveness.

**Limitations:**

The experimental results are unconvincing and require clear clarification. Specifically, the authors compare the proposed method and M-FedAvg with $\alpha=0.5$, aiming to verify the point "not-degraded". However, the performance of M-FedAvg does not align with previous works. For instance, M-FedAvg can achieve good test accuracy when $\alpha=0.5$, e.g., $>85\%$. This significantly weakens the contribution of this work. The authors must claim or clarify that the code base is the same for all experiments.

**Suitability:**

2

---

### Official Review · Reviewer_gkr6 · 2024-05-30

**Rating:** 3
**Confidence:** 3

**Summary:**

Summary:
This paper studies the one-shot federated learning problem where the communication between server and clients is only conducted one time. Due to existing methodologies that aggregate while partially discarding local knowledge in clients’ models would lead to significant performance degradation, the authors claim it is caused by stressing
too much on finding a one-fit-all model. In order to solve this problem, the authors propose to conduct a mixture-of-experts mechanism to guarantee all local knowledge stays intact, thus avoiding performance degradation in one-shot federated learning. Moreover, the authors propose self-supervised training by generating data guided by approximating underlying distributions of local data and making distinct predictions among experts. Once the self-supervised training is done on the gating network, a one-shot MoE training is conducted to guide the learning of each expert.

**Strengths:**

Strengths:
* The motivation is clear and reasonable.
* This paper is well-organized and easy to follow.
* The experimental results are good, and the evaluation is extensive and sufficient.

**Limitations:**

Weaknesses:
* How did Figure 1 is generated? If it is manually plotted, the conclusion of “local models are discarded with knowledge distillation and parameter optimization” is less convincing.
* Why employing self-supervised learning is important for the one-shot federated learning problem? It seems that the motivation for employing data generation to boost learning performance is orthogonal to the motivation of this paper. Could there be any justification? If self-supervised training is important, how does this module influence general performance? Moreover, what does the generated data look like? Are there any visualizations?
* It seems that the proposed method still has slight performance degradation than SCAFFOLD, is there any explanation?
* The similarity and informativeness are two other incremental contributions. For a fair comparison, it is suggested to employ these two modules to a typical baseline method and then compare the performance.
* The datasets for evaluation are relatively small-scale, it is suggested to incorporate several typical larger federated learning benchmarks.
* How efficient is the proposed method? It is suggested to conduct a theoretical or experimental evaluation to further justify the efficiency of the proposed method compared to typical federated learning baseline methods, such as SCAFFOLD.
* Federated learning under OOD scenarios is a very practical research problem, how does the proposed method perform under OOD datasets? Could the authors provide several insights or justification?

**Suitability:**

2

---

### Meta-Review · Area_Chair_xF2X · 2024-06-30

**Recommendation:** Accept (Poster)
**Confidence:** 5

**Metareview:**

This work aims to improve the model's performance under the one-shot federated learning scenario.

Most reviewers acknowledge the originality of the work. Before the rebuttal, the reviewers had concerns about unclear experimental results, unclear claims, and expensive computation costs. After the rebuttal, all reviewers acknowledged that their concerns had been solved.

Currently, the ratings of all reviewers are positive. Therefore, the AC would like to accept this paper. The authors are suggested to carefully revise the paper based on the concerns and suggestions from reviewers in the camera-ready version.